# Characterization of Vagus Nerve Stimulation (VNS) Dose-Dependent Effects on EEG Power Spectrum and Synchronization

**DOI:** 10.3390/biomedicines12030557

**Published:** 2024-03-01

**Authors:** Enrique Germany Morrison, Venethia Danthine, Roberto Santalucia, Andrés Torres, Inci Cakiroglu, Antoine Nonclercq, Riëm El Tahry

**Affiliations:** 1Institute of NeuroScience (IoNS), Université Catholique de Louvain, 1200 Bruxelles, Belgium; venethia.danthine@uclouvain.be (V.D.); roberto.santalucia@uclouvain.be (R.S.); andres.torres@uclouvain.be (A.T.); inci.cakiroglu@uclouvain.be (I.C.); riem.eltahry@saintluc.uclouvain.be (R.E.T.); 2Walloon Excellence in Life Sciences and Biotechnology (WELBIO) Department, WEL Research Institute, Avenue Pasteur 6, 1300 Wavre, Belgium; 3Department of Pediatric Neurology, Cliniques Universitaires Saint Luc, 1200 Bruxelles, Belgium; 4Bio- Electro- and Mechanical Systems (BEAMS), Université Libre de Bruxelles, 1050 Bruxelles, Belgium; antoine.nonclercq@ulb.be; 5Department of Neurology, Cliniques Universitaires Saint Luc, 1200 Bruxelles, Belgium

**Keywords:** refractory epilepsy, VNS dose-dependency, EEG central biomarker, cortical desynchronization, Individual VNS-dosing, EEG spectral analysis

## Abstract

This study investigates the dose-dependent EEG effects of Vagus Nerve Stimulation (VNS) in patients with drug-resistant epilepsy. This research examines how varying VNS intensities impacts EEG power spectrum and synchronization in a cohort of 28 patients. Patients were categorized into responders, partial-responders, and non-responders based on seizure frequency reduction. The methods involved EEG recordings at incremental VNS intensities, followed by spectral and synchronization analysis. The results reveal significant changes in EEG power, particularly in the delta and beta bands across different intensities. Notably, responders exhibited distinct EEG changes compared to non-responders. Our study has found that VNS intensity significantly influences EEG power topographic allocation and brain desynchronization, suggesting the potential use of acute dose-dependent effects to personalized VNS therapy in the treatment of epilepsy. The findings underscore the importance of individualized VNS dosing for optimizing therapeutic outcomes and highlight the use of EEG metrics as an effective tool for monitoring and adjusting VNS parameters. These insights offer a new avenue for developing individualized VNS therapy strategies, enhancing treatment efficacy in epilepsy.

## 1. Introduction

Vagus Nerve Stimulation (VNS) has played a fundamental role in neuromodulatory interventions [1]. It has proven beneficial for patients with drug-resistant epilepsy (DRE) [2] who are non-eligible or have failed epilepsy surgery. Recent findings show an encouraging increase in the responder rate, with the most significant seizure frequency reduction observed within the first one to two years after starting therapy [3,4]. The efficacy of VNS stimulation increases with time after implantation, and earlier exposure to VNS improves prognosis [5], which may result from VNS-induced neuroplastic changes [6].

Important to VNS’s therapeutic efficacy is its ability to induce neural desynchronization [7]. This concept has its roots in early animal studies [8,9,10] and has been further solidified by EEG connectivity studies in humans [7,11,12,13,14]. Analyses of epileptic brain structures have revealed a tendency for synchronization, suggesting heightened connectivity in malfunctioning excitatory neuronal pathways [15,16]. Advanced tools, such as functional connectivity and network theory, have been valuable in uncovering these epileptic brain networks. Yet, translating this understanding of neural desynchronization into practical, patient-specific therapeutic strategies has been challenging. A primary obstacle in this matter is the absence of an acute central biomarker that indicates optimal afferent vagus nerve activation for achieving antiepileptic effects.

Many VNS studies have traditionally focused on the effects of clinically programmed VNS intensities [13,17,18,19]. They study the acute effects of VNS by comparing VNS ON and OFF states and classifying patients into ‘Responders’ and ‘Non-Responders’. This generalized approach, while insightful, may only capture part of the spectrum of individual patient responses to varied VNS intensities. Notably, these studies operate within set clinical parameters, potentially missing more nuanced effects. A specific focus on varying VNS intensity levels within the same patient is less common in the literature [20,21].

In contrast, our research aims to investigate the effects of VNS on EEG by using different intensities in both individual and group behaviors. By doing so, we hope to offer a more holistic understanding of VNS’s therapeutic potential and evolve to a personalized stimulation approach.

## 2. Materials and Methods

### 2.1. Patient Selection

Patients with epilepsy resistant to medication and undergoing VNS therapy were selected from the VNS follow-up registry at the Centre for Refractory Epilepsy, Saint Luc University Hospital in Brussels, Belgium. This study’s protocols were reviewed by the Ethics Committee of Saint-Luc Hospital (Approval No. 2018/07NOV/416). Participants provided informed consent before participating in any research activities. To be included in this study, participants had to: (i) be aged between 18 and 65; (ii) presenting with focal/multi-focal or generalized epilepsy including Lennox-Gastaut syndrome; (iii) have had a VNS device (DemiPulse Model 103, DemiPulse Duo Model 104, AspireHC Model 105, or AspireSR Model 106; manufactured by LivaNova, Inc., London, UK) implanted for a minimum of six months; (iv) have VNS electrode impedance within the range of 2 to 5 kOhm, verified on the day of examination. Individuals were excluded if they had: (i) any accompanying laryngeal disease or damage to the recurrent laryngeal nerve; (ii) reported significant VNS-related adverse effects, like intense dyspnea (grade III–IV) or acute pain in the neck or ear area. Included patients were ensured to have no recent status epilepticus requiring recent adaptation of anti-seizure medication nor use of any rescue medication on the day of the experiment. They had a stable anti-seizure medication regimen at least two weeks prior to the day of the experiment.

The sole criterion for the VNS device settings was that the clinically programmed intensity was between 0.75 and 2.5 mA. During experimentation, VNS settings were kept as clinically programmed according to the Livanova guidelines, except for intensity, which was acutely tested within 0.25 to 2.25 mA. The effectiveness of VNS treatment was evaluated using the most recent clinical follow-up report available at the time of experimentation.

### 2.2. EEG Recordings and Epoch Selection

Electroencephalographic data were acquired using a 64-channel EEG system (Biosemi B.V, Amsterdam, The Netherlands), adhering to the 10–20 placement convention at a 4096 Hz sampling rate. Furthermore, Laryngeal Motor Evoked Potentials (LMEPs) were captured using two electrodes placed horizontally at mid-line, adjacent to the larynx, allowing for the detection of stimulation artifacts from the corresponding implanted VNS electrodes [22].

For the duration of the recordings, patients remained in a resting state while awake. Figure 1 shows the stimulation protocol used for the experimental procedure. Prior to initiating the experiment, the routine VNS was deactivated. The VNS intensities were programmed to increase progressively, starting at 0.25 mA and incrementing by 0.25 mA until the clinical intensity plus an additional 0.25 mA was reached. Then, the VNS intensities were decreased progressively in the reverse order. During the experiment, a magnet was utilized every 45 s to initiate stimulation, and each intensity level was maintained for three consecutive stimulations. After three stimulations at a given intensity, the system was reprogrammed to the next intensity level. This protocol entailed the gradual escalation of stimulation intensities, starting from the base level of 0.25 mA to the ceiling (clinical intensity + 0.25 mA), followed by a decremental phase returning to the minimal 0.25 mA, again with a 0.25 mA step adjustment. The ‘Magnet ON’ period for each stimulation was consistently set at 14 s.

Once acquired, the recorded data were imported into the Letswave 7 toolbox (version: 7, UCLouvain, Brussels, Belgium) for initial preprocessing. The EEG data were re-referenced to the common average to ensure consistent and reliable spatial information across all channels. Subsequent data processing and analysis were carried out using a custom-developed script in MATLAB (version: R2023a, Mathworks, Natick, MA, USA). Figure 2 presents the data segmentation procedure and subsequent analyses. The 64 EEG channels underwent a bandpass filter ranging from 0.5 Hz to 70 Hz. VNS stimulation artifacts were precisely identified based on the concomitant VNS-induced LMEP recording channel. For each detected VNS “ON” period, a 10 s epoch was extracted to represent the stimulation active phase centered on the stimulation period (seconds 3 to 13). After this “ON” window, another 10 s epoch (between seconds 25 and 35) was derived to signify the “OFF” period between consecutive stimulations. The presence of interictal activity was not an exclusion criterion. Nevertheless, all 10 s epochs were visually reviewed, showing only 4% of interictal activity on average. Ten-second epochs were chosen based on previous literature that has analyzed the sensitivity of EEG-derived metrics for phase coupling-derived metrics and functional connectivity analyses [23,24]. Fraschini et al. reported the sensitivity of epoch length over several metrics, including the phase lag index. They suggested using epoch lengths of at least 6 s and to consider the types of coupling captured by the adopted metric. Due to constraints of the possible programmable stimulation ON periods of VNS devices, we opted for a 10 s epoch analysis within a 14 s ON time, allowing us to optimize the tradeoff between the measured stability and experimentation time. Each stimulation pulse at a specific intensity was delivered on average within 90 s via the magnet-mode manual trigger. In the worst-case scenario, for patients at 2.0 mA clinical setting, this involved 9 intensity steps, with a total stimulation time of about one and a half hours (considering intensity reprogramming time). Visual inspection was performed to assess the correct and accurate segmentation for the 10 s interval selection, manually adjusting the detection threshold on a patient-by-patient basis. As presented below, segmented “ON” and “OFF” periods were used in topographical spectral analyses and channel-wise synchronization metrics.

### 2.3. Topographical Band Power Analysis

For spectral decomposition, the VNS “ON” and “OFF” epochs corresponding to each stimulation intensity were parsed through a filtering stage using a zero-phase delay second-order Butterworth bandpass filter tuned for each frequency sub-band. Low and high cut-off frequencies of each sub-band of interest were as follows: delta (0.5 Hz to 4 Hz), theta (4 Hz to 8 Hz), alpha (8 Hz to 13 Hz), beta (13 Hz to 30 Hz), and gamma (30 Hz to 70 Hz). The band ratio for every electrode position was computed to facilitate normalized spectral power comparisons and ensure consistency during across-subjects pooling.

This computation quantifies each frequency band’s power relative to the EEG spectrum’s total power. Topographical maps were subsequently generated for each stimulation intensity and band. An average representation of every topographical map per band was compared between responders and non-responders.

The segmented 10 s windows of EEG during the ON phase of the stimulation under different intensities were used for computing the spectral power of each band and normalized for each electrode channel with respect to the broadband total energy. In this way, it is possible to account for the amount of power contributed by each band during the epoch in every channel. This normalized power ratio is then represented as a 64-element vector and can be displayed on a scalp topological map.

After computing the spectral power ratios for all epochs, they were averaged by intensity level to create an intensity-dependent topographical allocation of the proportions representing each EEG band activity of each subject. A two-sample high-dimensional test was applied to the data for each intensity level and band to determine if VNS intensity generates an acute effect on the power spectrum of the EEG bands. The data were grouped by response to therapy category, namely responders vs. non-responders.

### 2.4. EEG Dose-Dependent Desynchronization Analysis

Previous literature has shown that EEG synchronization using phase difference-based metrics, such as the weighted phase lag index (wPLI), is sensitive to the acute desynchronizing effects of VNS on EEG [18,25,26]. The wPLI accounts for the stability or consistency of phase differences across EEG epochs. The weighted aspect of this metric reduces the contribution of in-phase and counter-phase signals that are usually associated with volume conduction. The wPLI pair-wise matrices were computed for each intensity level and frequency band across epochs.

The wPLI OFF/ON ratio was computed to account for the desynchronization effect of each intensity level. A ratio above one expresses the desynchronization effects of the applied VNS stimulation. To keep the individuality of the patient therapeutical response with respect to the brain cortical synchronization, patient groups were pooled with respect to their own clinical intensity (namely C_i_). Individual characteristics can be obtained on a per-patient basis, computing the intensity difference (intensity Δ) between each patient’s clinical intensity and the intensity at which each patient shows a maximal desynchronizing acute effect. Partial-responders were grouped jointly with responders due to their shared clinical outcomes.

### 2.5. Statistical Analysis

All spectral matrices processed using MATLAB from spectral and synchronization analyses were formatted and exported to tables identifying subject ID, intensity level, frequency band, and response to therapy. They were processed using R studio (version: 2023.06.0, Posit Software, PBC).

For the spectral topographical allocation maps, we employed a high-dimensional two-sample statistical test [27] to compare the spectral power maps for each intensity from 0.25 mA to 2.25 mA. This statistical test is suitable for cases where the number of dimensions (64 channels) exceeds the number of samples (patients). This test will give information on a large scale if the spectral distributions at each intensity level are significantly different from the responders vs. non-responder groups.

Thereafter, to have a deeper understanding of whether the topographical allocations of normalized spectral power over the scalp are different between groups, a channel pair-wise comparison was performed using a Mann–Whitney–Wilcoxon signed rank test. Bonferroni correction was used to account for multiple comparisons over multiple channels and intensities.

For the EEG synchronization analyses, a Mann–Whitney–Wilcoxon signed rank test was also used to compare whole brain wPLI desynchronization ratio OFF/ON values on specific bands across responder and non-responder groups over subsequent intensity levels. Finally, the clinically programmed intensity against the intensity at which the maximum desynchronization effect is obtained for broadband was individually analyzed. Intensity Δ (clinically programmed intensity − wPLI intensity that maximizes their cortical desynchronization) was computed for each patient. Each patient was color-coded according to the therapy response. The Kruskal–Wallis H-test was performed to determine between-group differences.

## 3. Results

### 3.1. Patient Cohort Demographics and Clinical Information

In this prospective study, we evaluated a cohort of 28 epileptic patients undergoing VNS therapy. The clinical programmed intensities of the VNS administered to these patients ranged from 0.75 to 2.0 mA. The registered clinical information is presented in Table 1.

Based on the observed therapeutic outcomes, patients were categorized into three distinct groups:Responders (R, 13 individuals) with a seizure reduction rate greater than 50%;Partial-responders (PR, 4 individuals) with a seizure reduction rate between 30% and 50%;Non-responders (NR, 11 individuals) with a seizure reduction rate of less than 30%.

### 3.2. Topographical Band Power Analysis Results

The topological maps show significant differences for the low-frequency delta bands, in the lower intensities of 0.5 mA, 1.0 mA, and 1.25 mA and a significant difference in the alpha band for 1.25 mA and for the beta band for all the intensity ranges. The significance of *p* < 0.05 after the Benjamini–Hochberg correction is shown in Table 2.

The sequences of averaged epochs on subsequent stimulation intensity levels show a consistent regional band proportion activity for the delta, theta, alpha, and beta bands, as shown in Figure 3. For topographical power band allocation differences (refer to Figure 3), Bonferroni’s most conservative correction method was used to minimize the multiple comparison effects of 64 channel locations and nine intensity levels. This lowers the effective alpha threshold to 8.6 × 10^−5^. Slow-frequency delta waves are predominantly active in the frontal region. The theta band showed higher power allocations in the central and parieto-occipital regions. Alpha band power was strongly present in bilateral central-parietal regions. Lastly, beta band power was consistently allocated to the frontocentral regions. Also, it can be observed that the power spectral proportion of the beta band over the scalp electrodes has higher values in the responder’s group than in the non-responder’s group, being consistent with the high-dimensional whole-brain statistical difference shown in Table 2.

### 3.3. EEG Dose-Dependent Desynchronization Results

Phase synchronization analysis was performed, grouping the wPLI OFF/ON ratio over intensities and taking the clinical setting as reference intensity, as shown in Figure 4. R and PR were grouped together and compared against NR. Averaging patients relative to their clinically programmed intensity allows us to observe group behavior for the synchronizing or desynchronizing effects of the VNS-applied intensity. The wPLI OFF/ON ratio computes the relation of phase synchronization for the OFF period and ON period for a specific VNS intensity. When the wPLI values over these two periods are the same or close to each other, a ratio of 1 is expected. The value 1 is marked as the horizontal dashed black lines in the plots of Figure 4. When the phase synchronization computed by the wPLI ON is higher than the OFF period, the wPLI OFF/ON ratio takes values below 1, indicating a synchronization effect. On the other hand, when the ON period has lower wPLI values, the ratio takes values above 1, indicating a desynchronization effect. The R and PR groups show a desynchronizing acute effect around the clinically programmed intensity (Ci), marked with the vertical blue dashed line, whereas the NR group did not. The graphs also show that there are lower intensity values than Ci, which achieves a higher acute desynchronizing effect. The Mann–Whitney–Wilcoxon signed rank test showed no statistical difference for any intensity.

To illustrate the relation of each patient’s clinically programmed intensity with respect to the acute desynchronizing effect of VNS, broadband wPLI OFF/ON was used to compute the intensity at which the maximal desynchronization effect was reached. Figure 5a shows a scatter plot with a diagonal line indicating a trend or relationship between the clinical intensity and the broadband OFF/ON wPLI maximizing intensity. The points are color-coded, with red (NR), green (PR), and blue (R) dots. The identity diagonal lines represent the times when clinical intensity is equal to maximizing intensity. This means that patients over the line or close to it manifest the maximal desynchronization effects within the therapy. However, the data points that reside in the region above the line labeled “Understim.”, suggesting under-stimulation, indicate that the programmed clinical intensity setting is currently lower than the intensity at which the maximal desynchronization ratio was achieved. Data points in the region under the line labeled “Overstim.”, suggesting over-stimulation, imply lower intensities at which maximal desynchronization is achieved. The intensity difference between the clinically programmed intensity (Ci) and the maximal desynchronization intensity is computed as intensity Δ. The mean clinical intensity setting is described as (NR: 1.5 mA, PR: 1.43 mA, R: 1.48 mA), and the mean wPLI broadband maximizing intensity is described as (NR: 0.91 mA, PR: 0.93 mA, R: 1.21 mA).

Figure 5b displays three box plots, each representing the categories NR in red, PR in green, and R in blue. There is a trend in the data suggesting that R have a clinical setting closer to the maximal desynchronization intensity, and NR are the most distant to it. The Kruskal–Wallis H-test did not show significant differences between groups (*p*-value = 0.281). However, effect size calculation using mean and standard deviation values for intensity Δ on each group showed a Cohen’s d value of 0.6, meaning a medium effect size interaction. Looking at the data distribution of intensity Δ for each group, it can be observed that while responders are gathered closer to an intensity Δ of 0, PR mostly differs from 0.375 mA or more and NR with 0.5 mA and over.

## 4. Discussion

In the present study, we aim to characterize EEG dose-dependent effects to determine a central biomarker for VNS that could lead to personalized VNS titration. We studied acute changes by performing topographical scalp mappings over the different band allocations, spectral power distributions, and intensity-dependent cortical synchrony measures. Our main finding shows that rather low intensity values (0.75~1.25 mA) achieve the maximal desynchronizing EEG effect. In contrast, 1.625 mA was reported to correspond to the programmed intensity for which, in a population-based study, a generalized mixed model showed the highest probability of becoming a responder [28]. Our results in Figure 4 show a clear trend towards higher desynchronization values at clinical intensities for the responder’s group but not for non-responders. This suggests that there might be an optimal intensity at which the acute stimulation effects are maximized. Figure 6 illustrates the concept of possible implications for a personalized titration process, basing the VNS intensity parameter adjustments on objective brain EEG metrics. In humans, recent unpublished studies exploring the acute VNS-induced functional effects with optimized protocols of stimulation-locked BOLD fMRI parametrically found peak activations in the thalamus at intermediate levels of intensity, whereas excessive increases in intensity led to decreased patterns of BOLD fMRI activation in the same regions [29]; interestingly, microburst stimulation patterns elicited similar dose-dependent responses at the brainstem and cerebellar level [30].

A previous study involving 18 subjects with drug-resistant epilepsy found no differences in spectral activity between periods before, during, and after VNS stimulation [17]. However, they observed consistent network organization changes, particularly in slow EEG bands, using graph theory metrics like the Small World Index and Global Efficiency. Our study shows that spectral power modulation is effectively subtle, and it can be further noticed in terms of topographical allocation rather than just quantifying whole brain average spectral power [17]. Another study focusing on spectral changes in EEG following VNS implantation reported principal changes in the lower EEG frequency bands, specifically in the delta and theta ranges. The study noted a decrease in delta rhythm and an increase in theta rhythm in relation to clinical benefit, suggesting that changes in these frequency bands might be related to the therapeutic effects of VNS [31]. This was particularly evident in patients who demonstrated a significant clinical response to VNS. Our results aligned with these previous results and added that there seems to be a specific location of delta band reduction in the temporal-central regions that is more effective in responders. The theta band relates to a central activation in responders, which can also be observed in NR at lower stimulation amplitudes. However, it tends to fade away with higher currents, displacing the theta activity to parieto-occipital regions. Also, in the alpha band, a higher ratio of alpha activity in the left parieto-central region can be observed in R, whereas for NR, this is greatly decreased for intensities above 1.5 mA. Finally, the beta band shows a consistent topographical allocation on the fronto-central regions during VNS ON. However, R shows a higher beta power in this region compared to NR. Similar results were reported by Yokoyama et al. [32] and Ma et al. [18], where higher beta oscillations were correlated with better responses for VNS. Using EEG data, Bodin et al. investigated the impact of VNS on the synchronicity of interictal EEG rhythms in patients with refractory epilepsy. The study involved 19 patients with drug-resistant epilepsy, using the phase lag index (PLI) to estimate the level of synchronization between scalp EEG signals during the ON and OFF phases of VNS therapy. The results showed that responders had a lower global level of EEG synchronization across all frequencies compared to non-responders [33]. Also, a study by Sangare et al. included 35 patients who underwent scalp EEG with 21 channels. The researchers analyzed the synchronization in EEG time series using the phase lag index (PLI) and phase-locking value (PLV), and they also computed brain network models based on the graph theory. The study’s main findings were that VNS could modulate functional connectivity, with ON periods associated with a reduction in EEG time-series synchronization in delta, theta, and beta bands at the awake state. This desynchronization was observed only in responder patients and was statistically correlated with changes in seizure frequency [13]. Research on VNS’s impact on sleep EEG showed that responders to VNS therapy experienced stronger theta band desynchronization during sleep but not in wakefulness. This suggests that the therapeutic effects of VNS might differ according to the state of vigilance and is specific to specific EEG bands [14]. Previous analysis of EEG time-series synchronization in patients treated with VNS for drug-resistant epilepsy revealed that, among responders, there was a significant decrease in synchronization during VNS ON periods in the delta, theta, and beta bands. This correlation between VNS-induced EEG desynchronization and a decrease in seizure frequency supports the hypothesis that VNS’s antiseizure effect is mediated by changes in interictal functional connectivity [13]. Grouping patient’s phase synchronization by therapy outcome can be tricky as each patient responds at different intensities. For this reason, grouping intensities relative to clinically programmed intensity (Ci) rather than absolute intensity allows the synchronization effects to add up locally for all patients at a specific clinically programmed value. Indeed, several patients must consistently show desynchronization or synchronization behaviors for the wPLI OFF/ON ratio to have values above or below 1. Otherwise, effects cancel each other out averaging towards 1. Our results showed the average behaviors showed a clear trend toward higher desynchronization values at clinical intensities for the responder’s group but not for non-responders, for whom these higher desynchronization effects were observed at lower intensities (see Figure 4). This effect is present on all bands (broadband), but it is especially noticeable for theta and alpha bands. Nevertheless, the final effect of VNS may also be the consequence of repetitive acute desynchronization effects over longer periods of therapy, which might be responsible for achieving neuroplastic changes in the brain [6,34,35]. This suggests that the clinical benefit of VNS results from a combination of acute effects targeting the afferent fibers, resulting in a specific brain network activation plus additional long-term effects achieved by means of neuroplastic changes with respect to the acutely activated networks.

Only a few studies have performed limited experiments with varying VNS intensities (with very limited intensity steps per patient) and usually used other biomarkers of responses, such as pupil dilations, instead of EEG. Also, some animal studies in rats by Morrison, Pruitt, and their colleagues have demonstrated the effects of VNS on cortical plasticity, which like the antiepileptic, are crucially mediated by LC modulation and noradrenergic transmission [36]. This study shows an inverted-U dose-dependency where moderate intensities (0.8 mA) elicit significantly and reproducibly a greater rearrangement of cortical maps in rats compared to sham, 0.4 mA, and 1.6 mA stimulations [20,37,38]. Even though the comprehensive study on VNS-induced pupil dilation by Mridha et al. in rats did not show a similar profile of dose dependency, another dose-dependent relationship was observed, notably an exponential sigmoid, tending to saturation [39]. However, instead of inverted-U (local maxima), the obtained exponential sigmoid curve could be a limitation of the maximal amplitude of 0.9 mA used in the methods. Other dose-dependent studies in patients used low-, mid-, and high-stimulation intensities graded with respect to clinical intensity and effectively revealed an inverted U-curve relationship between intensity and mean pupil dilation response (PDR) [21]. This evidence shows that specific VNS intensities may play an important role in central modulatory effects. Even though our results did not show a clear inverted-U response curve in terms of synchronization, we align with previous findings that certain intensities exert a local maxima effect in cortical desynchronization. Patients in the R and PR groups were shown to be clinically treated nearer this maximizing intensity compared to NR.

Current clinical titration recommendations use scheduled programming for a rapid titration scheme [40,41], targeting between 1.5 and 2 mA in the shortest time possible, in agreement with findings that support the idea that earlier intervention with VNS may lead to better outcomes and that the duration of epilepsy is associated with the probability of improved response to VNS therapy. Fahoum et al., in multiple clinical studies, analyzed the impact of different VNS dose parameters [28], such as output current, pulse width, and signal frequency, on treatment efficacy and side effects. Their main findings suggest that a target output current near 1.625 mA and signal frequencies of 20, 25, or 30 Hz are recommended, considering that individual patients may require adjustments based on their unique circumstances. Our findings state that the range between 0.75 mA and 1.25 mA tends to be the most distinct in terms of topographical spectral maps (see Figure 3). Also, maximal cortical desynchronizing effects (see Figure 4 and Figure 5) are obtained at an average intensity of 1.05 mA (NR: 0.91 mA, PR: 0.93 mA, R: 1.21 mA). Further multicentric clinical trials such as “Optimization of the Parameters of Vagal Nerve Stimulation (OPSTIMVAG)” registered at ClinicalTrials.gov under the number NCT04693221 will further provide evidence if a personalized dose based on maximal desynchronization (lowest phase lag index value) is effectively more efficient in obtaining positive clinical outcomes. Furthermore, these findings lay the groundwork for future research, potentially leading to the development of optimized, patient-specific VNS therapies.

Regarding the study’s limitations, we acknowledge that this study is based on a single recording session, with a heterogeneous therapy duration, but which was at least six months for all included patients. Also, patients presented differences in pulse width and chronic duty-cycle programmed for their therapy, which might impact neuroplasticity effects. This could affect the acute responses, opening new avenues for longitudinal studies to characterize how VNS intensity modulates brain regions and desynchronization over time. This work encourages future longitudinal studies that might help elucidate the relationship between acute and long-term chronic effects and neuroplasticity. In addition, we acknowledge that interictal activity might have biased our results. Nevertheless, the total duration of interictal activity compared to the epochs of this study was neglectable (only 4% on average). Also, the presence of a few interictal discharges highlights the potential clinical applicability of our study. At last, due to the heterogeneous population, we could not answer the question of whether the effect of VNS dose on EEG was dependent on the localization of the epilepsy.

## 5. Conclusions

These findings collectively highlight the critical role of individualized VNS dosing in optimizing therapeutic outcomes for patients with refractory epilepsy. They also emphasize the potential of EEG metrics as a tool for monitoring and adjusting VNS parameters to achieve the best possible clinical response, thereby reinforcing the significance of personalized medicine in the context of epilepsy treatment. Aligning clinical titration practices with reliable target dose biomarkers could significantly improve the outcomes of VNS treatment.

## Figures and Tables

**Figure 1 biomedicines-12-00557-f001:**
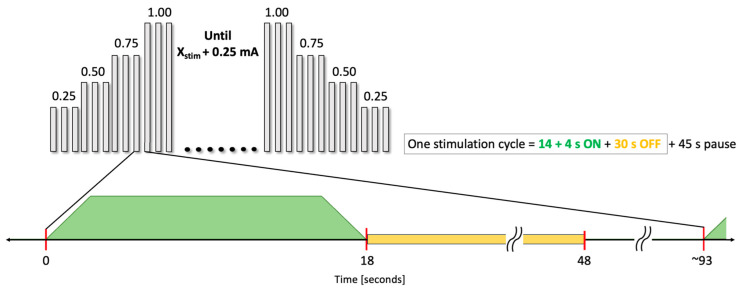
Stimulation protocol for experimental procedure. VNS pulses of 14 s ON with 2 s ramp-up and ramp-down are manually triggered using the magnet mode on the VNS device, having a 30 s OFF plus 45 s pause, leaving ~75 s between stimulation pulses.

**Figure 2 biomedicines-12-00557-f002:**
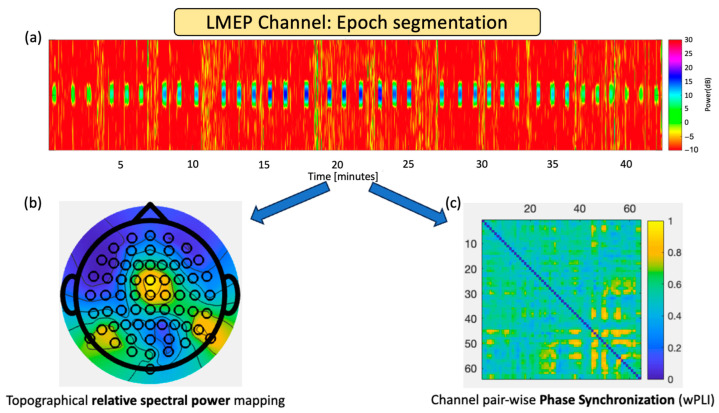
Data segmentation procedure and subsequent analyses. (**a**) Using the LMEP channel, stimulation epochs from the implanted VNS pulse generator are detected by searching harmonics artifacts from stimulation pulses. Segmented “ON” and “OFF” periods for each EEG channel are labeled according to stimulation intensity and used in (**b**) topographical spectral analyses and (**c**) channel-wise synchronization metrics.

**Figure 3 biomedicines-12-00557-f003:**
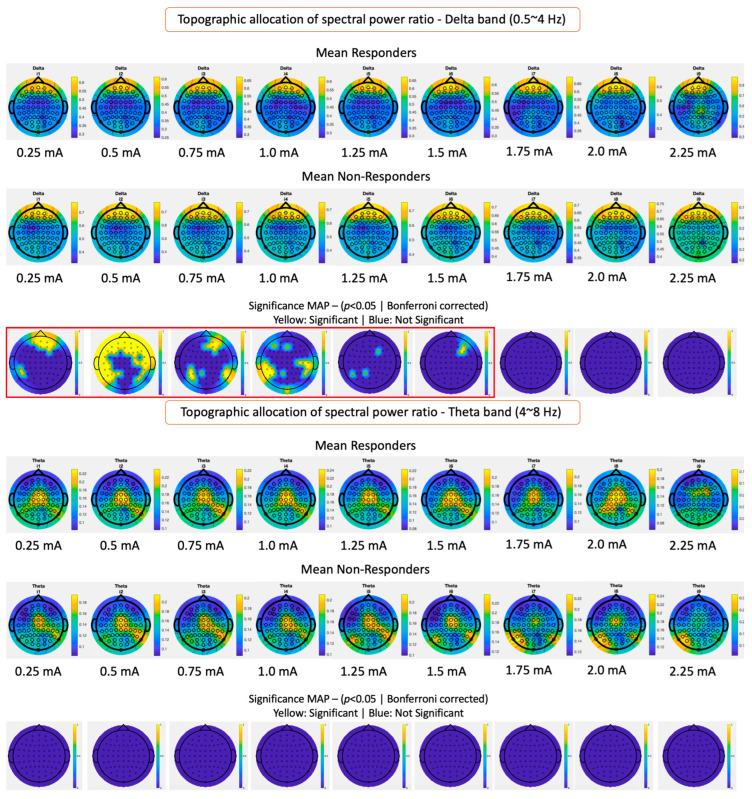
Topographical allocation maps of spectral power ratio over stimulation intensities for delta, theta, alpha, and beta bands. Average power band ratio maps are shown for responders and non-responders over increasing intensities; channel-wise comparison between groups was evaluated by the Mann–Whitney–Wilcoxon signed rank test, and significantly different electrodes are displayed in the topological map after Bonferroni correction.

**Figure 4 biomedicines-12-00557-f004:**
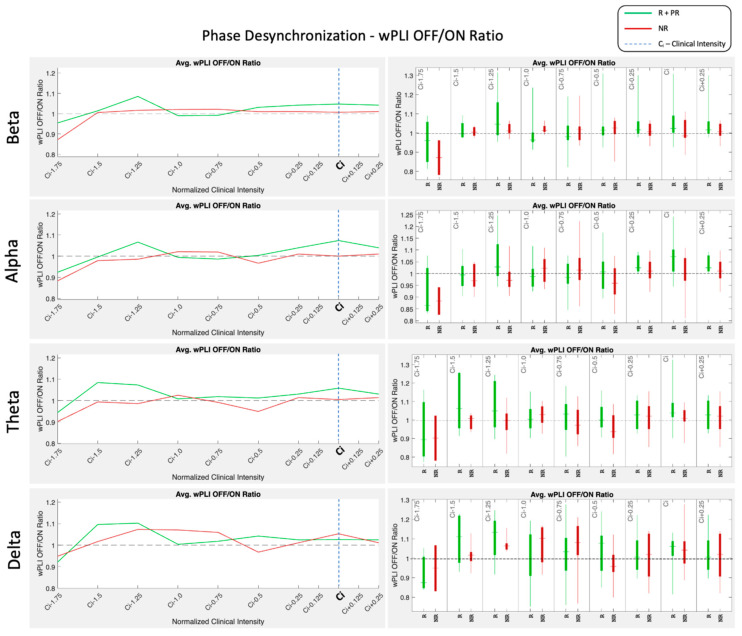
Phase desynchronization ratio wPLI OFF/ON for R + PR (green) and NR (red) organized by clinical intensity values for delta, theta, alpha, and beta bands.

**Figure 5 biomedicines-12-00557-f005:**
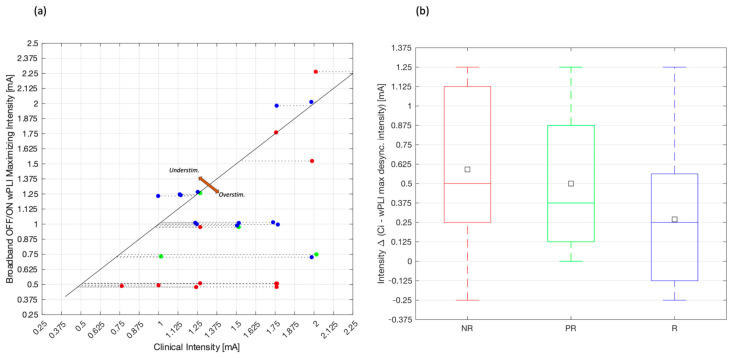
(**a**) Individual representation of patient clinical intensity setting vs. intensity at which the broadband wPLI OFF/ON desynchronization ratio is maximized. (**b**) Distribution of intensity difference from clinical intensity to broadband wPLI OFF/ON desynchronizing intensity. Points and boxplots are color-coded, with red (non-responders), blue (responders), and green (partial-responders) dots.

**Figure 6 biomedicines-12-00557-f006:**
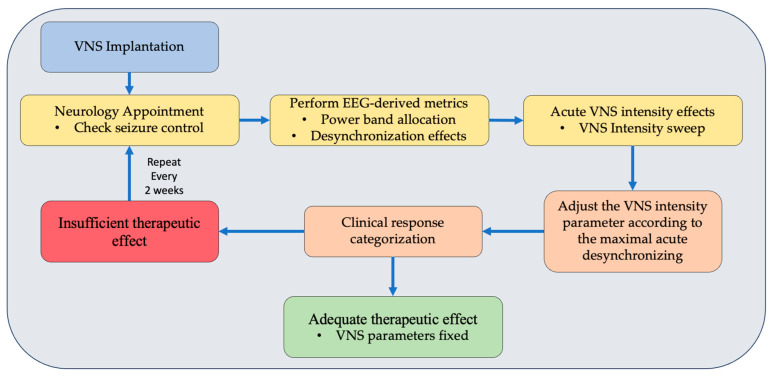
Concept diagram illustrating hypothetical implications of study findings. A personalized titration process using EEG-derived metrics of acute desynchronization effects of VNS intensity. Under this paradigm, VNS intensity can be adjusted up or down to match the maximal desynchronizing effects instead of rapidly targeting 1.75 mA under current guidelines.

**Table 1 biomedicines-12-00557-t001:** Patient demographics and clinical data.

Subject ID	Sex	Age Epilepsy Onset	Epilepsy Type	Clinical VNS Output Current [mA]	Clinical VNS Frequency [Hz]	Clinical VNS Pulse Width [us]	Response to Therapy Category
1	F	7	Idiopathic generalized	1.75	25	250	NR
2	M	25	Focal, unknown etiology	1	30	250	PR
3	M	15	Unknown	1.25	30	250	R
4	M	18	Focal	1.5	25	250	PR
5	M	36	Focal, structural	1.5	25	250	R
6	F	1	Idiopathic generalized	2	20	250	R
7	M	30	Focal, structural	1.25	20	250	NR
8	F	8	Focal, unknown etiology	1.25	20	250	R
9	F	16	Focal, unknown etiology	1.25	20	250	NR
10	F	14	Idiopathic generalized	1.125	30	250	R
11	M	18	Focal, structural	2	25	250	NR
12	F	12	Focal, genetic	1.25	20	250	NR
13	F	13	Focal, unknown etiology	1.75	20	250	R
14	F	28	Focal, structural	1	30	500	NR
15	F	36	Focal, unknown etiology	2	20	250	NR
16	F	5	Plurifocal, unknown etiology + structural	1.75	20	250	R
17	M	8	Bi-focal, structural	1	20	250	R
18	M	6	Plurifocal, unknown etiology	2	30	250	PR
19	M	9	Focal, unknown etiology	1.125	20	250	R
20	F	20	Focal, unknown etiology	1.25	20	250	PR
21	F	20	Focal, structural	1.75	20	250	NR
22	F		Idiopathic generalized	2	25	250	R
23	M	25	Focal, unknown etiology	0.75	25	250	NR
24	F	23	Focal, auto-immune	1.5	30	250	R
25	M	45	Focal, structural	1.75	30	250	NR
26	M	8	Focal, structural	1.75	30	250	R
27	F	15	Idiopathic generalized	1.75	20	250	NR
28	M	13	Idiopathic generalized	1.25	30	500	R

**Table 2 biomedicines-12-00557-t002:** Significance levels between R and NR topological maps for different stimulation intensity levels (*: *p* < 0.05 (Benjamini-Hochberg corrected).

VNS Intensity	Delta	Theta	Alpha	Beta	Gamma
0.25 mA	0.1110	0.7138	0.3424	0.0003 *	0.7696
0.5 mA	0.0005 *	0.7892	0.0520	<0.0001 *	0.7466
0.75 mA	0.1165	0.7944	0.1312	0.0027 *	0.7988
1.0 mA	0.0083 *	0.7575	0.2035	<0.0001 *	0.7415
1.25 mA	0.0136 *	0.8008	0.0029 *	0.0003 *	0.7929
1.5 mA	0.0998	0.7789	0.3677	<0.0001 *	0.5589
1.75 mA	0.6534	0.7554	0.7299	<0.0001 *	0.7329
2.0 mA	0.4272	0.7727	0.7580	0.0013 *	0.2258

## Data Availability

The data cannot be made publicly available upon publication because they are owned by a third party and the terms of use prevent public distribution. The data supporting this study’s findings are available upon reasonable request from the authors.

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
