# Peer review of "Characterization of Vagus Nerve Stimulation (VNS) Dose-Dependent Effects on EEG Power Spectrum and Synchronization"

_biomedicines, 2024, doi:10.3390/biomedicines12030557_

Round 1

Reviewer 1 Report

Comments and Suggestions for Authors

This study investigates the dose-dependent EEG effects of VNS in 28 patients with drug-resistant epilepsy. The methods involve EEG recordings at incremental VNS intensities on awake patients and subsequent spectral and synchronization analyses. Key findings show significant changes in EEG power in the delta, theta, alpha, and beta bands across different VNS intensities and distinct patterns between responders and non-responders, suggesting the potential of personalized VNS dosing for epilepsy treatment optimization.

Comments:

1. Details are lacking on inclusion/exclusion criteria related to epilepsy type and VNS device settings. Clarify this. 

2. Explain why 10-second epochs were chosen to represent ON/OFF phases. What impact could different durations have?

3. Validation is needed that the segmentation approach accurately captures stimulation artifacts for precise ON/OFF epoch detection. 

4. More details on the filter characteristics need be provided for the spectral decomposition.

5. Were medications kept constant for patients throughout the study? Clarify if any changes could impact the EEG responses.

6. How was the clinical VNS intensity selected for each patient? What programming guide or criteria was followed?

Author Response

2.14.0.0

Reviewer 2 Report

Comments and Suggestions for Authors

I reviewed the manuscript titled "Characterization of Vagus Nerve Stimulation (VNS) Dose-Dependent Effects on EEG Power Spectrum and Synchronization" submitted by Morrison et al. to the journal 'Biomedicines.' The study investigates the dose-dependent effects of Vagus Nerve Stimulation (VNS) on EEG patterns in patients with drug-resistant epilepsy, aiming to understand how varying VNS intensities impact EEG power spectrum and synchronization. The abstract provides a concise summary of the study's objectives, methods, and key findings. The categorization of patients into responders, partial responders, and non-responders based on seizure frequency reduction adds depth to the analysis. The study's conclusion emphasizes the significance of individualized VNS dosing for optimizing therapeutic outcomes in epilepsy treatment.

Consider adding a concept diagram to visually illustrate the main findings and the impact of varying VNS intensities on EEG patterns. This could enhance the understanding of the study's results for readers.

Some sentences could be revised for clarity and conciseness. For example, "The study concludes that VNS intensity significantly influences EEG patterns" could be refined to enhance readability.

I recommend minor revisions to improve language clarity and consider the addition of a concept diagram to enhance the manuscript's overall impact and readability.

Comments on the Quality of English Language

Moderate revisions required

Author Response

2.14.0.0

Reviewer 3 Report

Comments and Suggestions for Authors

Dear colleagues.

It has been a pleasure to review the present manuscript.

I have some comments:

- Table 1: this is results section, not material and methods.

- The division of patients based on their response to treatment, should also be results, not material and methods.

- Please, try to offer any kind of calculation about the statistical power of the present analysis based on the desgin and the population recruited.

- Think about reducing the number of tables and / or figures.

- A paragraph of limitations at the end of the discussion section should be included.

Author Response

2.14.0.0

Reviewer 4 Report

Comments and Suggestions for Authors

Dear Authors,

the manuscript is clear in its conclusions, less so in its methods. In particular, nothing is said about the paroxysmal activity (epileptiform graphelements) potentially present in the EEG segments analyzed. Were only segments free of paroxysms selected? In addition to the variations in background activity, was the effect that different stimulation intensities had on the paroxysms also examined?

Furthermore, did the topographic analysis of the power bands take into account, in focal epilepsies, the location of the epileptogenic area?

Some clarifications in this regard are necessary in order to make the conclusions of your study more reliable.

Reference [27]: check authors' names; add the link to the AES website.

Best regards.

Author Response

2.14.0.0

Round 2

Reviewer 3 Report

Comments and Suggestions for Authors

Reviewer 4 Report

Comments and Suggestions for Authors

Dear Editor, Dear Authors,

I appreciate the work that has been done to make the method used better understandable. I have no further comments.

Best regards,